# Adaptive Policy with Wait-$k$ Model for Simultaneous Translation

**Libo Zhao**[1,4*], **Kai Fan**[2], **Wei Luo**[2], **Jing Wu**[2], **Shushu Wang**[3],
**Ziqian Zeng**[1†], **Zhongqiang Huang**[2†]

[1]Shien-Ming Wu School of Intelligent Engineering, South China University of Technology
[2]Alibaba DAMO Academy, [3]ZheJiang University
[4]Department of Computing, Hong Kong Polytechnic University
wilbzhao@mail.scut.edu.cn, wangshushu0213@zju.edu.cn, zqzeng@scut.edu.cn
{k.fan,muzhuo.lw,wj334275,z.huang}@alibaba-inc.com

## Abstract

Simultaneous machine translation (SiMT) requires a robust read/write (R/W) policy in conjunction with a high-quality translation model. Traditional methods rely on either a fixed wait-$k$ policy coupled with a standalone wait-$k$ translation model, or an adaptive policy jointly trained with the translation model. In this study, we propose a more flexible approach by decoupling the adaptive policy model from the translation model. Our motivation stems from the observation that a standalone multi-path wait-$k$ model performs competitively with adaptive policies utilized in state-of-the-art SiMT approaches. Specifically, we introduce DaP, a divergence-based adaptive policy, that makes read/write decisions for any translation model based on the potential divergence in translation distributions resulting from future information. DaP extends a frozen wait-$k$ model with lightweight parameters, and is both memory and computation efficient. Experimental results across various benchmarks demonstrate that our approach offers an improved trade-off between translation accuracy and latency, outperforming strong baselines. [1]

## 1 Introduction

Simultaneous Machine Translation (SiMT) (Gu et al., 2017) poses a unique challenge as it generates target tokens in real-time while consuming streaming source tokens, mostly applying to the scenario of speech translation (Zhang et al., 2019, 2023; Fu et al., 2023). Unlike traditional machine translation (MT) (Bahdanau et al., 2015; Vaswani et al., 2017) where the entire source is available, SiMT requires a read/write (R/W) policy to determine whether to generate target tokens or wait for additional source tokens, along with the ability to trans-

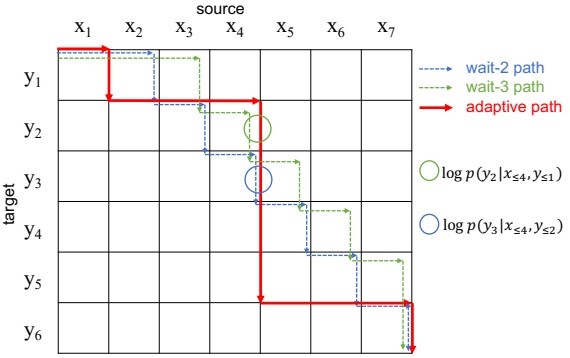

Figure 1: An example demonstrating that the optimization of cross-entropy loss on an adaptive path can be achieved through multi-path wait-$k$ training. Note that any adaptive path can be composed of subpaths (highlighted by the green and blue circles) derived from various wait-$k$ paths.

late from source prefixes to target prefixes (P2P) (Ma et al., 2018). Conventionally, the read/write policy and the translation model are designed to work in tandem: a simple wait-$k$ policy with an offline or wait-$k$ translation model (Ma et al., 2018; Elbayad et al., 2020; Zhang et al., 2021b), or an adaptive policy (Gu et al., 2017; Dalvi et al., 2018; Zheng et al., 2019, 2020; Ma et al., 2020a; Guo et al., 2023) that dynamically makes read/write decisions based on context, paired with a translation model that learns to translate the prefixes determined by the policy. The latter approach has achieved state-of-the-art results (Zhang and Feng, 2022, 2023). However, it entails dedicated architecture designs and multitask learning to jointly train tightly coupled adaptive policy and the translation model in order to balance translation quality and latency, resulting in computational complexity and challenges in optimizing individual components.

On the other hand, the multi-path wait-$k$ approach proposed by (Elbayad et al., 2020) introduces an effective method for training prefix-to-prefix translation models by randomly sampling dif-

---

[*] Work was done during Libo Zhao's research internship at DAMO Academy, Alibaba Group.
[†] Corresponding author.
[1]The code is available at https://github.com/lbzhao970/DaP-SiMT

ferent $k$ values between batches. Intuitively, as depicted in Figure 1, any read/write path determined by an adaptive policy can be composed of sub-paths from various wait-$k$ paths with different $k$ values, which can be effectively translated by a well-trained multi-path wait-$k$ model. Although the performance of the multi-path wait-$k$ approach falls behind the adaptive counterparts, we argue that this discrepancy should be attributed to the wait-$k$ policy rather than the translation model. Notably, our observations indicate that multi-path wait-$k$ models can achieve competitive results when combined with the adaptive policy proposed in (Zhang and Feng, 2022). This suggests a decoupled modular approach where the adaptive read/write policy can be modeled and optimized separately from the translation model, offering increased flexibility and the potential for improved performance.

A key aspect of this approach lies in acquiring high-quality signals to effectively supervise the learning of the read/write policy model. We draw inspiration from human simultaneous translation (Al-Khanji et al., 2000; Liu, 2008), where interpreters make a switch from listening to translating once they have gathered enough source context $\mathbf{x}_{\leq g(t)}$ to determine how to expand the partial translation $\mathbf{y}_{<t}$ to produce the next target word $y_t$. In other words, they anticipate that seeing additional future words would not impact their current decisions. This behavior implies a small discrepancy between the interpreters' modeling of translation distribution given the partial source context $p(y_t|\mathbf{y}_{<t}, \mathbf{x}_{\leq g(t)})$, and the translation distribution given the full source context $p(y_t|\mathbf{y}_{<t}, \mathbf{x})$. Conversely, interpreters would wait for more source words if the discrepancy becomes significant.

This observation motivates the utilization of statistical divergence (Lee, 1999) between the two conditional distributions for any prefix-to-prefix pair given a translation model as an informative criterion for making read/write decisions. In light of this, we propose DaP-SiMT, a novel divergence-based adaptive policy for simultaneous translation, to enable adaptive simultaneous translation using estimated divergence values, considering that the full source context is unavailable during the translation process.

While there are various options of neural architectures for the policy model and the translation model, we choose to build upon a well-trained multi-path wait-$k$ translation model with frozen parameters, and introduce additional lightweight parameters for the adaptive policy model. This design choice minimizes the memory and computation overhead introduced by the policy model, while providing an effective mechanism to achieve an adaptive read/write policy within an existing SiMT model. Our main contributions can be summarized as follows.

1. We propose a novel method to construct read/write supervision signals from a parallel training corpus based on statistical divergence.

2. We present a lightweight policy model that is both memory and computation efficient and enables adaptive read/write decision-making for a well-trained multi-path wait-$k$ translation model.

3. Experiments conducted on multiple benchmarks demonstrate that our approach outperforms strong baselines and achieves a superior accuracy-latency trade-off.

## 2 Related Works

Existing SiMT policies are mainly classified into fixed and adaptive categories. Fixed policies (Ma et al., 2018; Elbayad et al., 2020; Zhang et al., 2021b) determine read/write operations based on predefined rules. For example, the wait-$k$ policy (Ma et al., 2018) first reads $k$ source tokens and then alternates between writing and reading one token. On the other hand, adaptive policies predict read/write operations based on the current source and target prefix, achieving a better balance between latency and translation quality. Reinforcement learning has been used by (Gu et al., 2017) to learn the policy within a Neural Machine Translation (NMT) model. Dalvi et al. (2018) designed an incremental decoding that outputs a varying number of target tokens. Meanwhile, Arivazhagan et al. (2019) and Ma et al. (2020a) presented approaches to learning the adaptive policy through attention mechanisms. Recent advancements like the wait-info policy (Zhang et al., 2022b) and ITST (Zhang and Feng, 2022) have quantified the waiting latency and information weight respectively to devise adaptive policies. To the best of our knowledge, ITST is currently the state-of-the-art method in SiMT.

One of the most relevant works to ours is the Meaningful Unit (MU) for simultaneous translation (Zhang et al., 2020). This approach detects whether the translation of a sequence of source

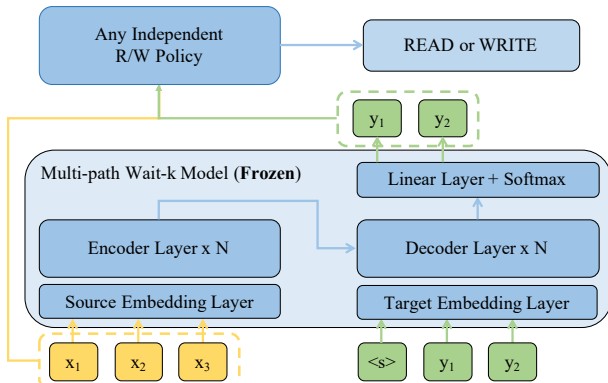

Figure 2: The combination of multi-path wait-$k$ model with any independent read/write policy model for simultaneous translation inference.

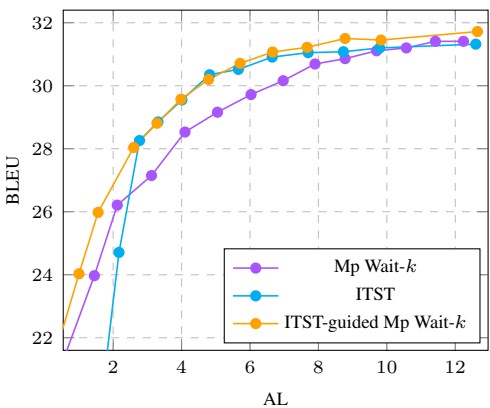

Figure 3: Comparison of BLEU vs. AL curves between multi-path (abbreviated as Mp) wait-$k$, ITST, and ITST-guided multi-path wait-$k$.

tokens forms a prefix of the full sentence's translation. This method was further generalized to speech translation in MU-ST (Zhang et al., 2022a). While MU-ST inspects the target prefix in the vocabulary domain, our work advances further by examining the distribution of target tokens.

## 3 Preliminary

### 3.1 Full-sentence MT and SiMT

In the context of a full sentence translation task, given a translation pair $\mathbf{x} = (x_1, x_2, ..., x_N)$ and $\mathbf{y} = (y_1, y_2, ..., y_T)$, an encoder-decoder model such as Transformer (Vaswani et al., 2017) maps $\mathbf{x}$ into hidden representations and then autoregressively decodes the target tokens. Generally, the model is optimized by minimizing the cross-entropy loss.

$$\mathcal{L}_{\mathrm{mt}} = -\sum_{t=1}^{T} \log p\left(y_t \mid \mathbf{x}, \mathbf{y}_{<t}\right) \quad (1)$$

For the SiMT task, given that $g(t)$ is a monotonic non-decreasing function representing the end timestamp of the source prefix that must be consumed to generate the $t$-th target token, the objective function of SiMT can be modified as follows,

$$\mathcal{L}_{\mathrm{simt}} = -\sum_{t=1}^{T} \log p\left(y_t \mid \mathbf{x}_{\leq g(t)}, \mathbf{y}_{<t}\right). \quad (2)$$

### 3.2 Wait-k Policy and Multi-Path Wait-k

**Wait-$k$ policy** (Ma et al., 2018), the most widely used fixed policy, begins by reading $k$ source tokens and then alternates between writing and reading one token. The function $g(t)$ for the wait-$k$ policy can be formally calculated as,

$$g(t; k) = \min\{t + k - 1, N\}. \quad (3)$$

**Multi-path Wait-$k$** (Elbayad et al., 2020) is an efficient technique for wait-$k$ training. It randomly samples different $k$ values between batches during model optimization. Concretely, the loss of one training batch is computed as follows,

$$k \sim \mathrm{Uniform}(\mathcal{K})$$
$$\mathcal{L}_{\mathrm{simt}}^{\mathrm{mp}} = -\sum_{t=1}^{T} \log p(y_t | \mathbf{x}_{\leq g(t;k)}, \mathbf{y}_{<t}), \quad (4)$$

where $\mathcal{K}$ is the candidate set of $k$. The main advantage of this method is its ability to make inferences under different latencies with a single model. Additionally, by adopting the unidirectional encoder, it can cache the encoder hidden states of streaming input. Previous experiments have shown that its performance is comparable to multiple wait-$k$ models trained with different $k$ values.

## 4 Method

### 4.1 Motivation

Typically, a fixed $k$ value is used when performing inference with a multi-path wait-$k$ model, following the wait-$k$ read/write path. As exemplified in Figure 1, any adaptive read/write path can be composed of subpaths of various wait-$k$ paths. Given that a multi-path wait-$k$ model is trained to perform prefix-to-prefix translation for different $k$ values, we argue that such a model can also achieve competitive performance when used with an adaptive read/write policy.

To evaluate this hypothesis, we construct a SiMT system by combining a multi-path wait-$k$ model with the adaptive policy from ITST (Zhang and Feng, 2022), as shown in Figure 2. Although ITST

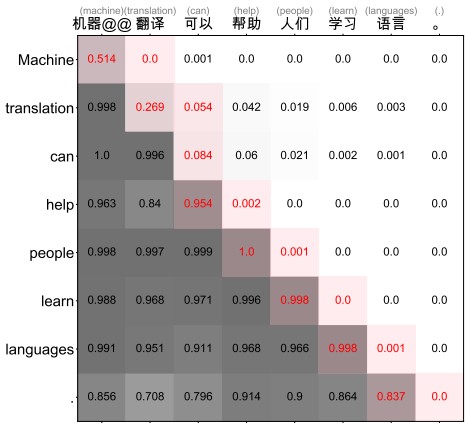

Figure 4: Example of a Zh→En divergence matrix $\mathbf{D}$ using cosine distance, where $\mathbf{D}_{t,g(t)} = \mathbf{D}\left(\mathbf{p}_t^{\text{part}}, \mathbf{p}_t^{\text{full}}\right)$. A potential read/write path is indicated by the red elements in the matrix and can be determined based on a predefined threshold (0.2 in this case).

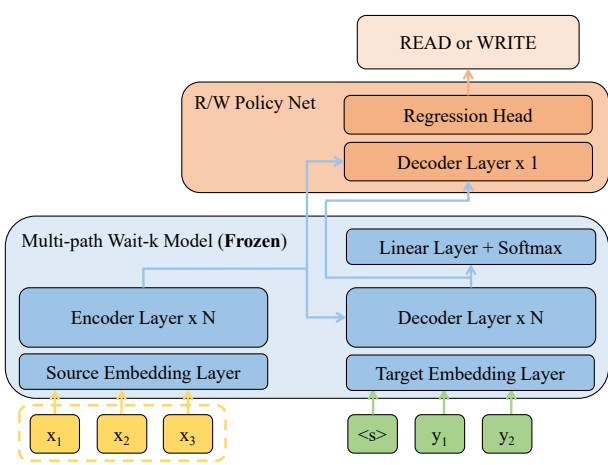

Figure 5: Architecture of the DaP-SiMT approach. It is equivalent to adding an extra decoder layer to the original SiMT. The output of the extra decoder will be fed into a regression head to determine read/write action.

has an integrated translation module, we only utilize its policy module to make read/write decisions and rely on the multi-path wait-$k$ model for translation. As illustrated in the BLEU/AL (BLEU Score vs. Average Lagging) curves in Figure 3, the resulting ITST-guided multi-path wait-$k$ approach achieves competitive performance in all latency settings. It consistently outperforms the original multi-path wait-$k$ approach. When compared to the original ITST approach, the combined approach achieves significantly better BLEU scores in the low latency region while obtaining comparable results in mid-to-high latency settings.

This positive observation leads to a natural question: Can we develop a better adaptive policy for a multi-path wait-$k$ model?

### 4.2 Divergence-based Read/Write Supervision

Learning an adaptive policy requires high-quality read/write supervision training data. It is unfeasible to collect manual annotations due to both the cost and complexity of the task. Instead, we draw inspiration from human simultaneous translation and propose to create such data automatically. We consider that a good write action should only occur when the partial source information is sufficient to make accurate translations, i.e., the translations should be similar to that with the complete source input. To be precise, we want to quantify the divergence $\mathbf{D}\left(\mathbf{p}_t^{\text{part}}, \mathbf{p}_t^{\text{full}}\right)$ between two distributions, one computed given partial source input and an-

other given full source:

$$\mathbf{p}_t^{\text{part}} = p(y_t = \cdot | \mathbf{x}_{\leq g(t)}, \mathbf{y}_{<t}) \quad (5)$$
$$\mathbf{p}_t^{\text{full}} = p(y_t = \cdot | \mathbf{x}, \mathbf{y}_{<t}), \quad (6)$$

where the distributions can be computed using a well-trained offline translation model or simultaneous translation model. In this paper, we quantify $\mathbf{D}(\cdot, \cdot)$ with three different divergence measures[2]:

$$\begin{aligned}
\text{Euclidean:} \quad & \|\mathbf{p}_t^{\text{part}} - \mathbf{p}_t^{\text{full}}\|_2 \\
\text{KL-divergence:} \quad & \text{KL}\left(\mathbf{p}_t^{\text{part}} \| \mathbf{p}_t^{\text{full}}\right) \\
\text{Cosine distance:} \quad & 1 - \cos\left(\mathbf{p}_t^{\text{part}}, \mathbf{p}_t^{\text{full}}\right)
\end{aligned}$$

We utilize the divergence measures to automatically construct read/write supervisions from a parallel corpus used for MT training. For each parallel sentence pair, we compute a divergence matrix $\mathbf{D}$, where each element $\mathbf{D}_{t,g(t)} = \mathbf{D}(\mathbf{p}_t^{\text{part}}, \mathbf{p}_t^{\text{full}})$, for all possible prefix-to-prefix pairs. We can then make read/write decisions by comparing $\mathbf{D}_{t,g(t)}$ with a threshold $\lambda$:

$$write \text{ if } \mathbf{D}_{t,g(t)} < \lambda, \text{else } read \quad (7)$$

Figure 4 shows an example divergence matrix and a highlighted read/write path. Varying the threshold would result in a different latency. For each chosen threshold, we could construct

---

[2]Although cosine distance does not meet the indiscernible condition ($d(a, b) = 0 \leftrightarrow a = b$) of statistical divergence for general vectors, it does for probability distribution vectors)

read/write samples and train a separate read/write policy model, however, the resulting model would be specific for that threshold. Instead, we treat the divergence measures computed from parallel sentences as ground-truth values, and train a single adaptive read/write policy model to predict the ground-truth values from the partial source and target pair:

$$\left(\mathbf{x}_{\leq g(t)}, \mathbf{y}_{<t}\right) \xrightarrow{\text{R/W Policy Net}} \mathbf{D}_{t,g(t)} \qquad (8)$$

### 4.3 SiMT with Adaptive Policy

The architecture of our proposed DaP-SiMT model, as depicted in Figure 5, integrates a SiMT model with a divergence-based adaptive read/write policy network. In contrast to the design shown in Figure 2, which employs an independent policy, our approach incorporates a policy network that leverages the hidden states from the SiMT model's encoder and decoder as inputs, tailored explicitly to enable adaptive read/write decisions within the SiMT model.

More specifically, the policy network includes an additional transformer decoder layer placed atop the original SiMT decoder, followed by a regression head responsible for predicting divergence values. The incorporation of an extra functional decoder layer aligns with the common practices found in previous works on NMT (Li et al., 2022, 2023). The design of the regression head adheres to Roberta (Liu et al., 2019), featuring two linear layers with a `tanh` activation function sandwiched in between. In terms of the learning objective, we employ Mean Squared Error (MSE) for divergence measures based on Euclidean distance or KL-divergence, and binary cross-entropy with continuous labels for measures based on cosine distance.

During training, we only tune the parameters of the adaptive policy network while keeping the parameters of the multi-path wait-$k$ model fixed. In the inference phase, we compare the predicted divergence values with a predefined threshold to make read/write decisions, following Equation 7, and can achieve varying latency levels by adjusting the threshold. Additionally, we have empirically observed that introducing another hyper-parameter to limit the maximum number of continuous READ operations for certain languages (see analysis in Section 5.4.2 for the impact on different language pairs) results in a better balance between transla-

tion quality and latency. The inference process of DaP-SiMT is summarized in Algorithm D in the Appendix.

## 5 Experiments

### 5.1 Datasets

**WMT2022 Zh→En**[3]. We use a subset with 25M sentence pairs for training[4]. We first tokenize the Chinese and English data using the Jieba Chinese Segmentation Tool[5] and Moses[6], respectively, and then apply BPE with 32,000 merge operations. We employ a validation set of 956 sentence pairs from BSTC (Zhang et al., 2021a) as the test set.

**WMT15 De→En**[7]. All 4.5M sentence pairs from this dataset are used for training, and are tokenized using 32K BPE merge operations. We use newstest2013 (3000 sentence pairs) for validation and report results on newstest2015 (2169 sentence pairs).

**IWSLT15 En→Vi**[8]. All 133K sentence pairs from this dataset (Luong and Manning, 2015) are used for training. We use TED tst2012 (1553 sentence pairs) for validation and TED tst2013 (1268 sentence pairs) as the test set. Following the settings in (Ma et al., 2020b), we adopt word-level tokenization and replace rare tokens (frequency $< 5$) with <unk>. The vocabulary sizes are 17K for English and 7.7K for Vietnamese, respectively.

### 5.2 Settings

All our implementations are based on the Transformer (Vaswani et al., 2017) architecture and adapted from the Fairseq Library (Ott et al., 2019). For the Zh→En experiments, we utilize the transformer big architecture, while the base and small architectures are used for De→En and En→Vi experiments respectively. We use the cosine distance to calculate the read/write supervision signals in the main experiments, and investigate the effects of different divergence types in Section 5.4.2.

For evaluation, following ITST (Zhang and Feng, 2022), we report case-insensitive BLEU (Papineni et al., 2002) scores to assess translation quality and Average Lagging (AL/token) (Ma et al., 2018) to measure latency. Regarding the maximum num-

---

[3] www.statmt.org/wmt22

[4] The data sources include casia2015, casict2011, casict2015, datum2015, datum2017, neu2017, News Commentary V16, ParaCrawl V9.

[5] https://github.com/fxsjy/jieba

[6] https://github.com/moses-smt

[7] www.statmt.org/wmt15

[8] nlp.stanford.edu/projects/nmt

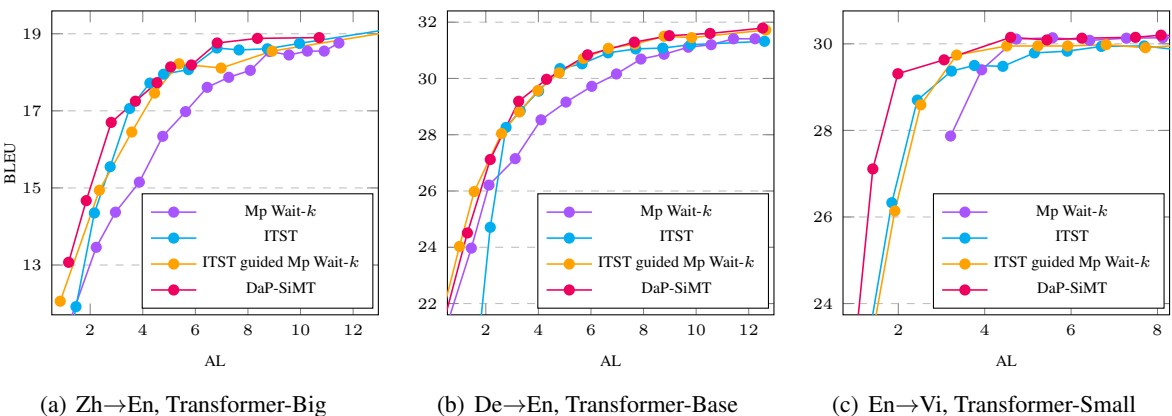

(a) Zh→En, Transformer-Big     (b) De→En, Transformer-Base     (c) En→Vi, Transformer-Small

Figure 6: Comparison of BLUE vs. AL curves between multi-path (abbreviated as Mp) wait-k, ITST, ITST-guided multi-path wait-k, and our proposed DaP-SiMT approach on three language pairs.

ber of continuous read actions in our method, we empirically select the best-performing configurations, which are no constraint, 4, no constraint for Zh→En, De→En, En→Vi respectively.

### 5.3 Main Results

The performance of our method is compared to previous approaches on three language pairs in Figure 6.

First, it is evident that the performance of a multi-path wait-$k$ model can be significantly improved when guided by an adaptive read/write policy like ITST, compared to using a fixed wait-$k$ policy. This enhanced performance often closely matches or even surpasses that of ITST, the previous state-of-the-art SiMT model, particularly in low latency settings for De→En translation. These results underscore the competitiveness and flexibility of prefix-to-prefix translation within the multi-path wait-$k$ model, a potential that remains largely untapped with a fixed wait-$k$ policy.

Secondly, our proposed DaP-SiMT approach significantly enhances the performance of the multi-path wait-$k$ model, outperforming all other approaches. The divergence-based adaptive policy consistently surpasses the fixed wait-$k$ policy across all latency levels. Furthermore, when compared to the adaptive policy in ITST, it achieves comparable results in the De→En scenario and superior performance in the Zh→En and En→Vi translations, all while using the same multi-path wait-$k$ model as the translation model. This result suggests that the divergence-based approach not only surpasses the fixed wait-$k$ policy but also competes effectively with state-of-the-art approaches

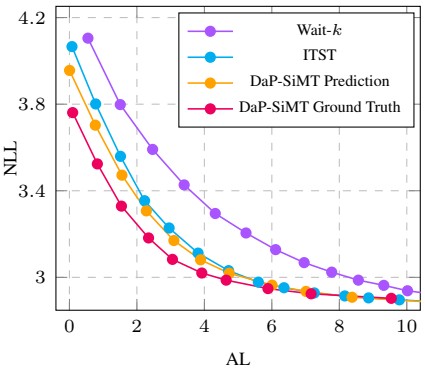

Figure 7: NLL vs. AL curves comparing four read/write policies utilizing the same SiMT model. "DaP-SiMT Ground Truth" indicates read/write paths derived from ground truth divergence values calculated using the full sentence, while "DaP-SiMT Prediction" is based on divergence values predicted by the policy model.

like ITST, which features a closely integrated adaptive policy and translation model.

### 5.4 Analysis

In our analysis, we aim to provide a more in-depth understanding of our proposed approach. Unless otherwise stated, results are based on the Zh→En Transformer-Big model.

#### 5.4.1 The NLL vs. AL Curve

In addition to the commonly used BLEU vs. AL curves that assess the quality of complete translations across different latency levels, we introduce a novel evaluation metric: the NLL vs. AL curve. This metric enables a qualitative measurement of the average impact of various read/write policies on translation quality at each translation step, all while utilizing the same translation model. To construct

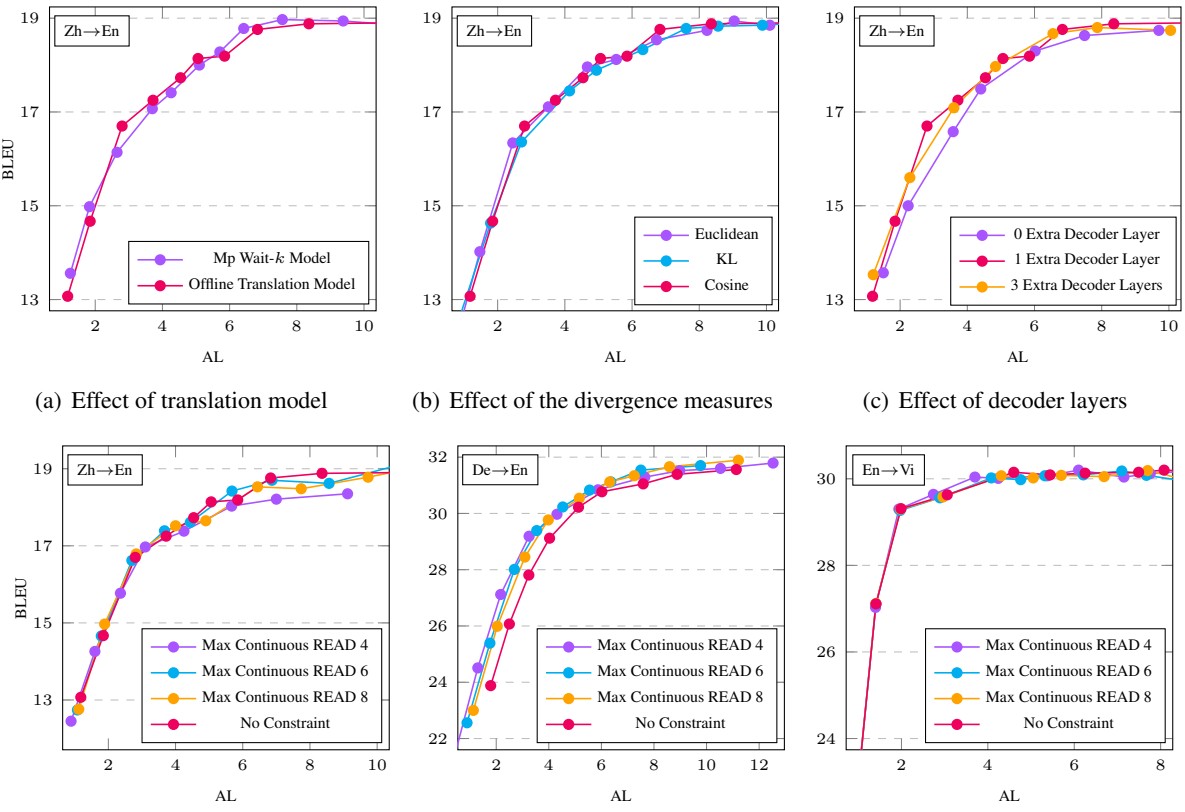

(a) Effect of translation model

(b) Effect of the divergence measures

(c) Effect of decoder layers

(d) Effect of the constraint on the maximum number of continuous read

Figure 8: Ablation studies on the proposed DaP-SiMT method

the NLL vs. AL curve, we begin with a read/write policy and the necessary hyper-parameters, such as $k$ for the fixed wait-$k$ approach and the threshold $\lambda$ for the divergence-based adaptive policy, which controls the latency level. Given a parallel sentence, we first derive the read/write path, denoted as $g(1), g(2), ..., g(T)$, under the read/write policy, and then calculate the negative log-likelihood of the translation along the read/write path, following Eq. (2). By aggregating these NLL scores and their corresponding latency levels across an entire dataset, we can generate NLL vs. AL curves for any read/write policy.

Figure 7 provides a comparative analysis of NLL vs. AL curves for four read/write policies: wait-$k$, ITST, and two variations of DaP-SiMT. The first DaP-SiMT variant is based on divergence values predicted by the policy model, while the second relies on ground truth divergence values computed using full sentences. The results underscore the effectiveness of our DaP-SiMT approach, as it consistently yields substantially lower NLL scores when compared to the fixed wait-$k$ policy at equivalent latency levels. This confirms that the SiMT

model is more adept at accurately predicting the correct translation along the resulting read/write paths. Furthermore, the DaP-SiMT approach exhibits a lower NLL curve compared to ITST's policy model, aligning with the trends observed in the BLEU vs. AL curves depicted in Figure 6. Notably, the DaP-SiMT variant employing ground truth divergence values has a much lower curve than its counterpart based on predicted divergence values. This suggests that there is potential for further improvement through better modeling of divergence supervision signals.

### 5.4.2 Ablation Study

**Effect of translation model for divergence supervision** In our main experiment, we utilized an offline translation model to calculate the divergence supervision values. Here, we assess the impact of utilizing a SiMT model for this purpose. Figure 8(a) illustrates that supervision signals computed by the multi-path wait-$k$ model result in almost identical performance to those obtained from the offline model.

**Effect of the divergence measures** We exam-

ine the influence of various divergence measures, including Euclidean distance, KL-divergence, and cosine distance, on the performance of DaP-SiMT. As depicted in Figure 8(b), the almost identical curves suggest that our approach is not sensitive to the choice of divergence measures.

**Effect of the number of layers for the policy net**   To examine whether a single additional decoder layer can effectively model the divergence supervision signals, we conducted comparative experiments using either 0 or 3 additional decoder layers. Figure 8(c) demonstrates that configurations with 1 or 3 additional decoder layers yield similar results. Although the configuration with 0 additional decoder layers does not perform as strongly, it still manages to achieve a reasonable balance between accuracy and latency.

**Effect of the max continuous READ constraint**   As discussed in Section 4.3, we introduce a constraint on the maximum number of continuous reads during inference, forcing a write action after reaching the limit. Figure 8(d) shows that this constraint has varying impacts on different language pairs. In the cases of Zh→En and En→Vi, this hyperparameter has minimal influence on results, especially in the low-latency region, which is a primary focus in SiMT. However, for De→En, a substantial improvement is observed with the introduction of this constraint.

We hypothesize that this difference is related to the modeling difficulty for language pairs with varying degrees of word order variations. As quantified in Appendix B, the De→En translation direction exhibits the highest anticipation rate among the three language pairs and demonstrates the most significant divergence in word order (Wang et al., 2023). Consequently, it naturally requires more read actions for an accurate De→En translation, which is reflected in the distribution of divergence supervision signals and subsequently influences the learned policy model. By adjusting the threshold $\lambda$ to achieve low latency, we inadvertently exacerbate the negative effects of exposure bias, resulting in excessive reads, as observed in our experiments. Introducing the maximum number of continuous reads serves as an ad-hoc solution to address this challenge, and we leave it to future research to investigate this issue thoroughly.

### 5.4.3   Upper Bound of DaP-SiMT

We evaluate the upper bound performance of DaP-SiMT to study the impact of modeling errors within

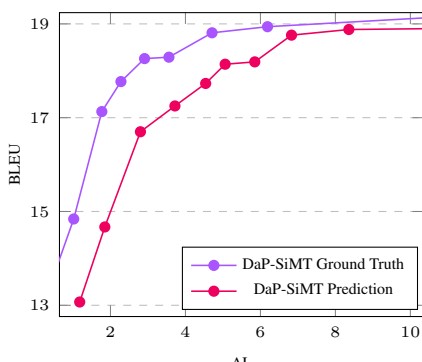

Figure 9: BLEU vs. AL curves comparing between DaP-SiMT with ground truth divergence and standard DaP-SiMT with predicted divergence.

the policy model. Specifically, we substitute the model-predicted divergence scores with the ground truth divergence calculated using the complete sentence $\mathbf{D}\left(\mathbf{p}_t^{\text{part}}, \mathbf{p}_t^{\text{full}}\right)$ during DaP-SiMT inference, following the procedure outlined in Algorithm D. As illustrated in Figure 9, in line with the findings from Section 5.4.1, the upper bound performance of DaP-SiMT is substantially higher than that achieved with the learned policy model. This observation highlights the potential for further improvement in policy modeling.

### 5.4.4   Examples

We provide several examples in Appendix A comparing the divergence matrix predicted by the learned policy model with the ground truth. Although there are some discrepancies, the predicted divergence matrix closely resembles the ground truth matrix and can be used to make reasonable read/write decisions.

## 6   Conclusion

In this paper, we introduce a divergence-based adaptive policy for SiMT, which makes read/write decisions based on the potential divergence in translation distributions resulting from future information. Our approach extends a frozen multi-path wait-$k$ translation model with lightweight parameters for the policy model, making it memory and computation efficient. Experimental results across various benchmarks demonstrate that our approach provides an improved trade-off between translation accuracy and latency compared to strong baselines. We hope that our approach can inspire a novel perspective on simultaneous translation.

## Limitations

Our evaluation primarily focused on assessing the impact of the proposed adaptive policy on simultaneous translation using BLEU vs. AL and NLL vs. AL curves. However, we acknowledge that intrinsic evaluations of the policy model itself are lacking, and further investigation in this area is necessary to guide improvements. We provided only a limited exploration of modeling variations for the policy model, leaving room for more in-depth analysis and enhancements. It's worth noting that while the threshold parameter $\lambda$ controls latency, it doesn't have a direct one-to-one relationship with latency, as is the case with the fixed wait-$k$ policy. This nuanced aspect requires careful consideration in future investigations.

## Ethics Statement

After careful review, to the best of our knowledge, we have not violated the ACL Ethics Policy.

## Acknowledgements

We would like to thank all the anonymous reviewers for the insightful and helpful comments. This work was supported by Guangzhou Basic and Applied Basic Research Foundation (2023A04J1687), and Alibaba Group through Alibaba Research Intern Program.

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

## A  Divergence Matrix Examples

Divergence matrix examples are shown in Figure 10, Figure 11, Figure 12.

## B  Anticipation Rate

Anticipation occurs in simultaneous translation when a target word is generated prior to the receipt of its corresponding source word. To detect instances of anticipation, accurate word alignment between the paired sentences is needed. Fast-align (Dyer et al., 2013) is employed to obtain the word alignment $a$ between a source sentence $\mathbf{X}$ and a target sentence $\mathbf{Y}$. The resulting alignments comprise a set of source-target word index pairs $(s, t)$, where the $s^{\text{th}}$ source word $x_s$ aligns with the $t^{\text{th}}$ target word $y_t$. A target word $y_t$ is $k$-anticipated ($A_k(t, a) = 1$) if it aligns to at least one source word $x_s$ where $s \geq t + k$:

$$A_k(t, a) = \mathbb{1}[\{(s, t) \in a | s \geq t + k\} \neq \varnothing]$$

The $k$-anticipation rate ($AR_k$) of an ($\mathbf{X}$, $\mathbf{Y}$, a) triple is further defined as follow:

$$AR_k(\mathbf{X}, \mathbf{Y}, a) = \frac{1}{|\mathbf{Y}|} \sum_{t=1}^{|\mathbf{Y}|} A_k(t, a)$$

The anticipation rates (AR%) of different language pairs are shown in Table 2, from which we can find that the De→En translation task exhibits greater word order differences than the other two cases.

## C  How Sensitive Is The AL To Thresholds During Inference?

Table 1 exhibits the sensitivity of the AL to thresholds during inference based on different settings of divergence types. It can be observed that the AL is not particularly sensitive to the threshold overall, which makes the process of determining the threshold straightforward.

## D  Algorithm

The inference process of DaP-SiMT is summarized in Algorithm 1.

## E  Numerical Results

The numerical results are presented in Table 3, Table 4, Table 5.

| *Divergence Type* | | | | | |
| Euclidean Distance | | KL Divergence | | Cosine Distance | |
| AL | Threshold | AL | Threshold | AL | Threshold |
|---|---|---|---|---|---|
| 0.67 | 0.5 | 0.72 | 1.8 | 1.18 | 0.52 |
| 1.47 | 0.4 | 1.37 | 1.4 | 1.85 | 0.4 |
| 2.1 | 0.33 | 2.21 | 1.0 | 2.8 | 0.26 |
| 3.51 | 0.24 | 3.03 | 0.7 | 3.72 | 0.18 |
| 4.67 | 0.2 | 4.07 | 0.5 | 4.54 | 0.14 |
| 5.53 | 0.18 | 4.94 | 0.4 | 5.85 | 0.1 |
| 6.73 | 0.16 | 6.32 | 0.3 | 6.83 | 0.08 |
| 7.42 | 0.15 | 7.61 | 0.24 | 8.36 | 0.06 |
| 9.04 | 0.13 | 8.56 | 0.2 | 10.71 | 0.04 |
| 10.01 | 0.12 | 9.88 | 0.16 | | |

Table 1: The sensitivity of the AL to the threshold during inference based on different settings of divergence types.

| Experiment | $k = 1$ | $k = 3$ | $k = 5$ | $k = 7$ |
|---|---|---|---|---|
| De→En | 30.4 | 15.2 | 8.5 | 5.1 |
| Zh→En | 25.4 | 12 | 6.3 | 3.6 |
| En→Vi | 17.3 | 5.2 | 1.9 | 0.8 |

Table 2: Anticipation rates (AR%) of different language pairs

---

**Algorithm 1:** SiMT inference with DaP

**Input:** streaming source tokens: $\mathbf{X}_{\leq j}$,
        threshold: $\delta$,
        target idx: $i \leftarrow 1$,
        source idx: $j \leftarrow 1$,
        max continuous READ constraint:
        $r_{max}$,
        current number of continuous
        READ: $r_c \leftarrow 1$

**Output:** target tokens: $\mathbf{Y} \leftarrow \{\texttt{<BOS>}\}$

1 **while** $\mathbf{Y}_{i-1} \neq \texttt{<EOS>}$ **do**
2     calculate R/W confidence $c$ with $\mathbf{Y}_{i-1}$ using the R/W decision net mentioned in 4.3;
3     **if** $c \leq \delta$ or $r_c \geq r_{max}$ **then**
4         translate $y_i$ with $\mathbf{X}_{\leq j}, \mathbf{Y}_{\leq i-1}$;
5         **if** $y_i \neq \texttt{<EOS>}$ **or** $j \geq |\mathbf{X}|$ **then**
6             // execute WRITE action
7             $\mathbf{Y}$.Append($y_i$);
8             $r_c \leftarrow 0$;
9             $i \leftarrow i + 1$;
10         **else**
11             // execute READ action
12             $j \leftarrow j + 1$;
13             $r_c \leftarrow r_c + 1$;
14     **else**
15         // execute READ action
16         $j \leftarrow j + 1$;
17         $r_c \leftarrow r_c + 1$;
18 **return** $\mathbf{Y}$

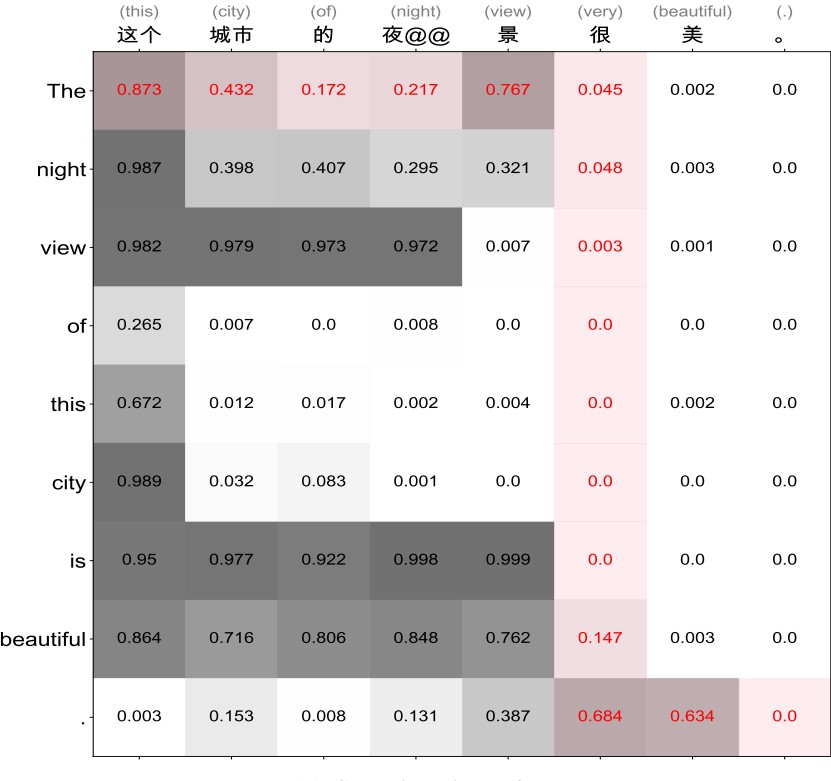

(a) Ground Truth Matrix

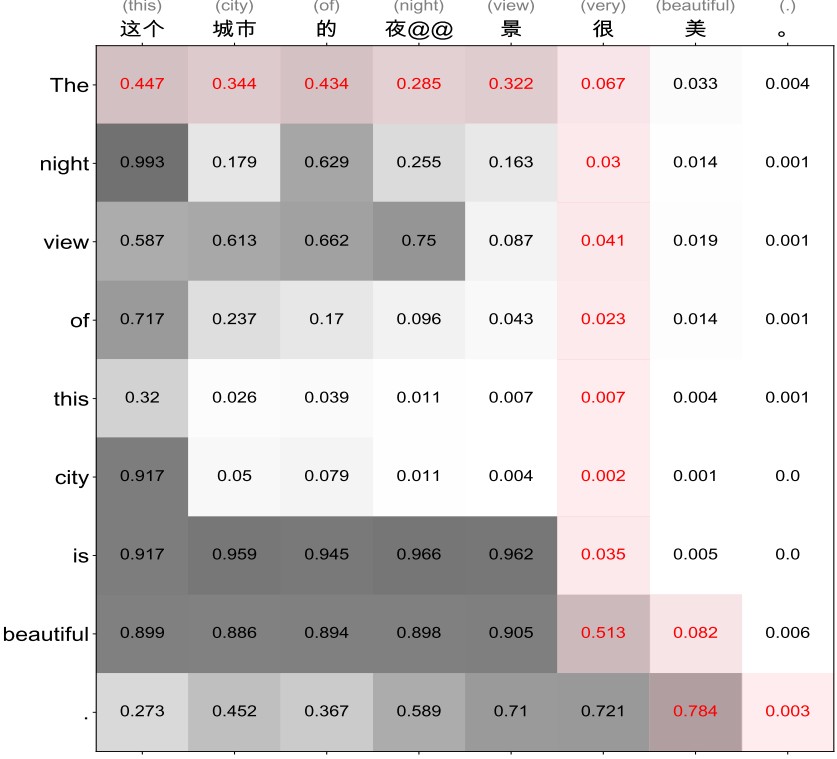

(b) Prediction Matrix

Figure 10: Example 1

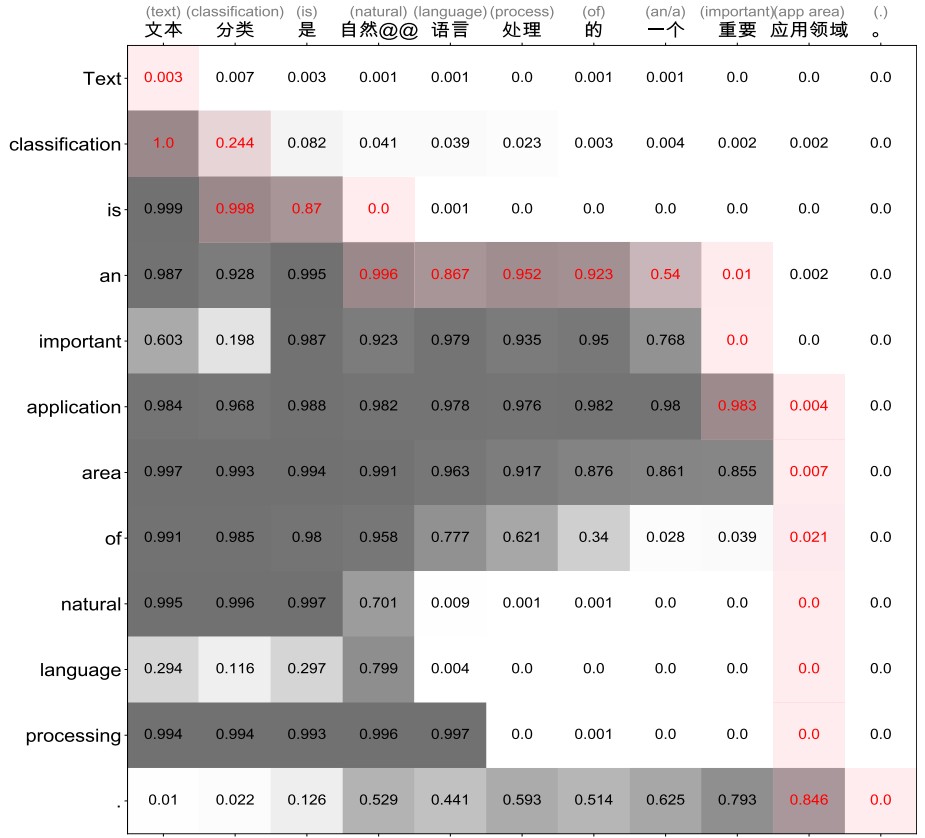

(a) Ground Truth Matrix

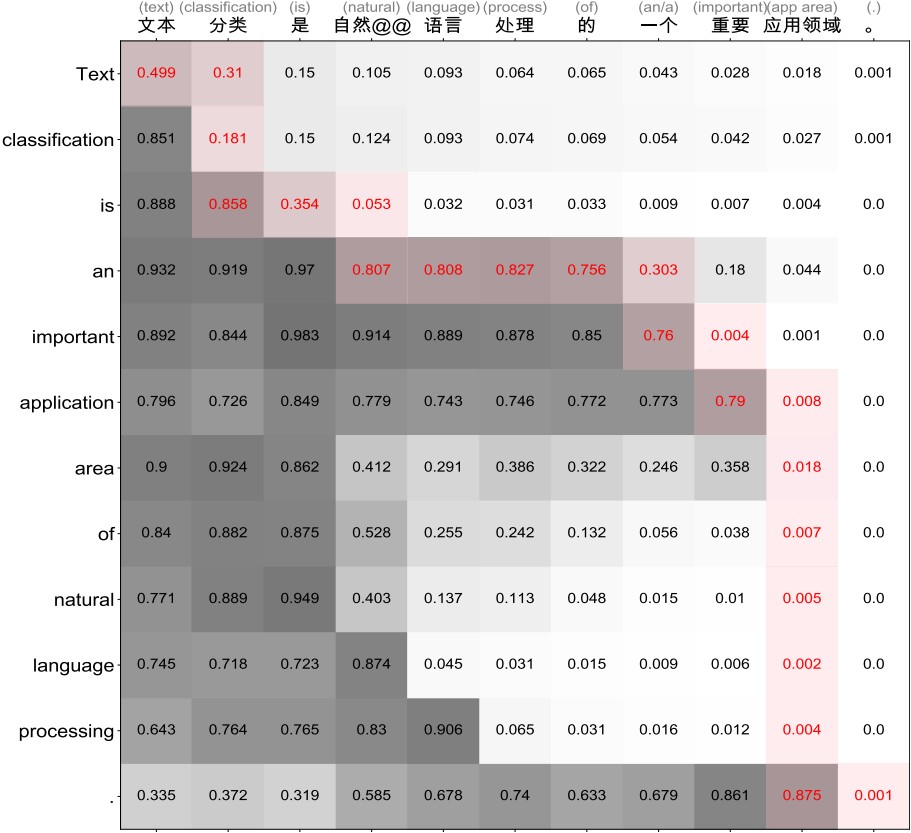

(b) Prediction Matrix

Figure 11: Example 2

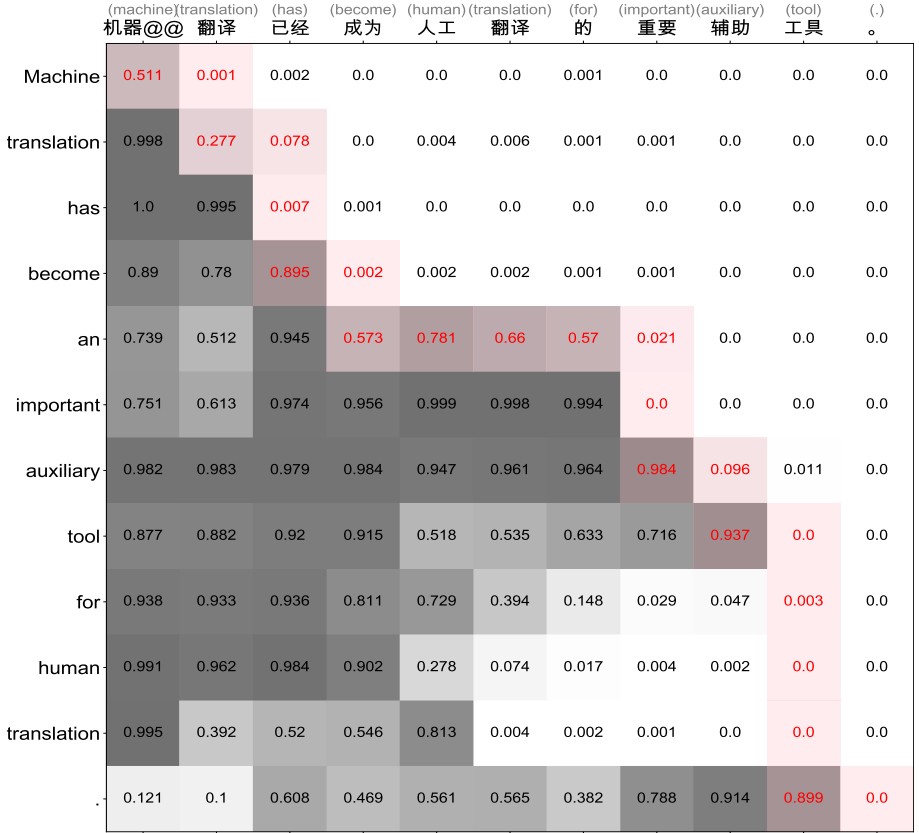

(a) Ground Truth Matrix

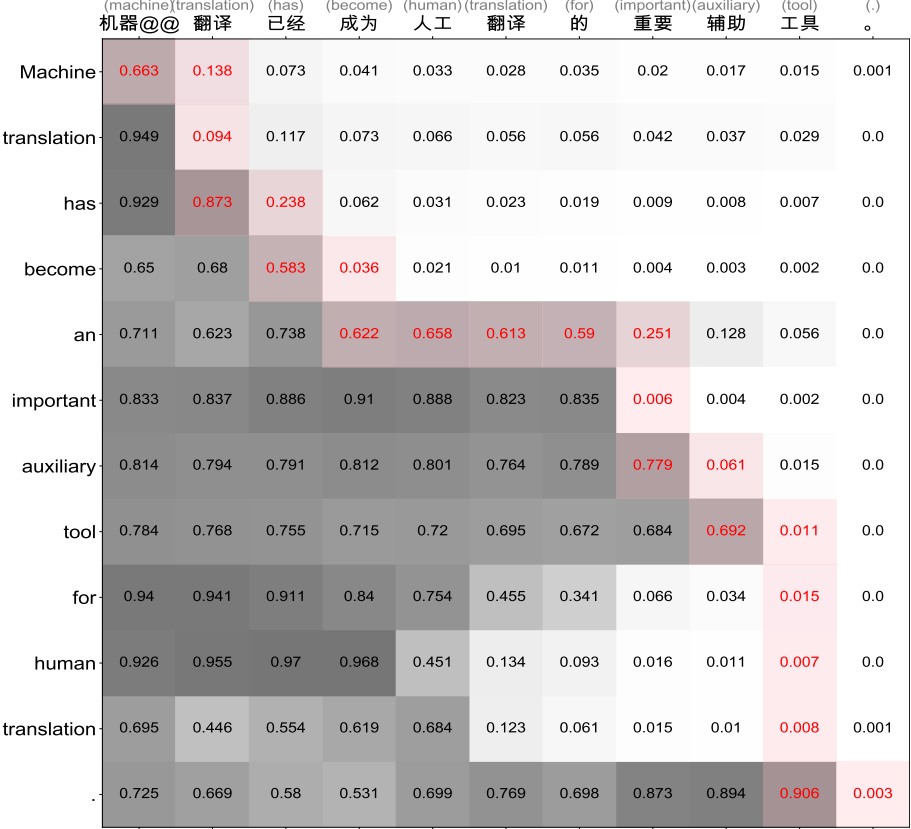

(b) Prediction Matrix

Figure 12: Example 3

**Main Results (Figure 6)**

**Zh→En**

| Mp Wait-k | | ITST | | ITST guided Mp Wait-k | | DaP-SiMT | |
|---|---|---|---|---|---|---|---|
| AL | BLEU | AL | BLEU | AL | BLEU | AL | BLEU |
| 1.31 | 11.7 | 0.7 | 8.91 | 0.86 | 12.06 | 1.18 | 13.07 |
| 2.23 | 13.46 | 1.46 | 11.92 | 2.35 | 14.94 | 1.85 | 14.67 |
| 2.96 | 14.37 | 2.16 | 14.35 | 3.58 | 16.45 | 2.8 | 16.7 |
| 3.87 | 15.15 | 2.76 | 15.55 | 4.45 | 17.46 | 3.72 | 17.25 |
| 4.76 | 16.34 | 3.5 | 17.06 | 5.38 | 18.22 | 4.54 | 17.73 |
| 5.63 | 16.98 | 4.27 | 17.72 | 6.98 | 18.11 | 5.06 | 18.14 |
| 6.45 | 17.61 | 4.79 | 17.95 | 8.92 | 18.55 | 5.85 | 18.19 |
| 7.27 | 17.87 | 5.74 | 18.07 | 13.52 | 19.06 | 6.83 | 18.76 |
| 8.09 | 18.05 | 6.82 | 18.63 | | | 8.36 | 18.88 |
| 8.82 | 18.54 | 7.66 | 18.58 | | | 10.71 | 18.9 |
| 9.56 | 18.45 | 8.74 | 18.61 | | | | |
| 10.26 | 18.55 | 9.96 | 18.75 | | | | |
| 10.9 | 18.55 | 13.68 | 19.15 | | | | |
| 11.46 | 18.76 | | | | | | |

**De→En**

| Mp Wait-k | | ITST | | ITST guided Mp Wait-k | | DaP-SiMT | |
|---|---|---|---|---|---|---|---|
| AL | BLEU | AL | BLEU | AL | BLEU | AL | BLEU |
| 0.47 | 21.08 | 1.57 | 19.2 | 0.49 | 22.15 | 0.49 | 21.65 |
| 1.45 | 23.97 | 2.17 | 24.71 | 1 | 24.03 | 1.3 | 24.51 |
| 2.12 | 26.21 | 2.77 | 28.26 | 1.56 | 25.98 | 2.17 | 27.12 |
| 3.12 | 27.15 | 3.31 | 28.85 | 2.6 | 28.04 | 3.25 | 29.19 |
| 4.1 | 28.53 | 4.01 | 29.55 | 3.28 | 28.81 | 4.31 | 29.97 |
| 5.05 | 29.16 | 4.82 | 30.35 | 3.98 | 29.57 | 5.87 | 30.84 |
| 6.03 | 29.72 | 5.66 | 30.52 | 4.79 | 30.2 | 7.65 | 31.29 |
| 6.97 | 30.16 | 6.65 | 30.91 | 5.71 | 30.71 | 8.98 | 31.52 |
| 7.9 | 30.69 | 7.7 | 31.05 | 6.66 | 31.07 | 10.53 | 31.6 |
| 8.78 | 30.86 | 8.73 | 31.08 | 7.67 | 31.22 | 12.53 | 31.79 |
| 9.7 | 31.11 | 9.79 | 31.2 | 8.78 | 31.5 | | |
| 10.57 | 31.2 | 12.6 | 31.32 | 9.83 | 31.45 | | |
| 11.42 | 31.41 | | | 12.65 | 31.72 | | |
| 12.24 | 31.41 | | | | | | |

**En→Vi**

| Mp Wait-k | | ITST | | ITST guided Mp Wait-k | | DaP-SiMT | |
|---|---|---|---|---|---|---|---|
| AL | BLEU | AL | BLEU | AL | BLEU | AL | BLEU |
| 3.21 | 27.87 | 1.29 | 23.06 | 1.3 | 22.71 | 0.89 | 21.89 |
| 3.93 | 29.4 | 1.85 | 26.33 | 1.92 | 26.14 | 1.41 | 27.11 |
| 4.73 | 30.11 | 2.44 | 28.7 | 2.52 | 28.59 | 1.99 | 29.31 |
| 5.57 | 30.14 | 3.23 | 29.37 | 3.35 | 29.74 | 3.06 | 29.63 |
| 6.43 | 30.08 | 3.76 | 29.5 | 4.51 | 29.95 | 4.6 | 30.15 |
| 7.28 | 30.13 | 4.42 | 29.48 | 5.23 | 29.95 | 5.44 | 30.09 |
| 8.12 | 30.14 | 5.15 | 29.79 | 5.92 | 29.95 | 6.25 | 30.13 |
| 8.93 | 30.11 | 5.91 | 29.83 | 6.81 | 29.98 | 7.49 | 30.15 |
| 9.7 | 30.1 | 6.7 | 29.94 | 7.72 | 29.91 | 8.08 | 30.2 |
| 10.43 | 30.2 | 7.69 | 29.95 | 8.71 | 29.98 | 8.74 | 30.17 |
| 11.13 | 30.16 | 8.67 | 29.84 | 9.95 | 30.07 | 9.61 | 30.01 |
| 11.79 | 30.13 | 9.93 | 29.95 | 12.55 | 30.09 | 10.67 | 30.11 |
| 12.41 | 30.16 | 12.58 | 30.01 | | | 11.69 | 30.1 |
| 13.01 | 30.18 | | | | | | |

**NLL vs. AL curves (Figure 7)**

**Zh→En**

| Wait-k | | ITST | | DaP-SiMT prediction | | DaP-SiMT Ground Truth | |
|---|---|---|---|---|---|---|---|
| AL | NLL | AL | NLL | AL | NLL | AL | NLL |
| 0.549 | 4.105 | 0.076 | 4.066 | 0.004 | 3.956 | 0.096 | 3.761 |
| 1.507 | 3.798 | 0.778 | 3.801 | 0.767 | 3.703 | 0.828 | 3.524 |
| 2.466 | 3.591 | 1.514 | 3.559 | 1.557 | 3.472 | 1.542 | 3.329 |
| 3.401 | 3.427 | 2.228 | 3.354 | 2.28 | 3.307 | 2.346 | 3.182 |
| 4.323 | 3.295 | 2.955 | 3.228 | 3.097 | 3.17 | 3.054 | 3.083 |
| 5.234 | 3.205 | 3.813 | 3.112 | 3.888 | 3.081 | 3.924 | 3.02 |
| 6.112 | 3.128 | 4.718 | 3.031 | 4.738 | 3.019 | 4.646 | 2.987 |
| 6.958 | 3.068 | 5.6 | 2.978 | 6.005 | 2.964 | 5.885 | 2.948 |
| 7.774 | 3.024 | 6.36 | 2.952 | 7.013 | 2.935 | 7.164 | 2.924 |
| 8.564 | 2.987 | 7.259 | 2.928 | 8.378 | 2.908 | 9.541 | 2.903 |
| 9.318 | 2.963 | 8.162 | 2.914 | 10.67 | 2.886 | | |
| 10.021 | 2.938 | 8.868 | 2.905 | | | | |
| 11.276 | 2.912 | 9.779 | 2.896 | | | | |
| | | 11.027 | 2.885 | | | | |

Table 3: Numerical results in Figure 6 and Figure 7.

**Translation Model Type (Figure 8(a))**

| | Mp Wait-$k$ Model | | Offlien model | |
|---|---|---|---|---|
| | AL | BLEU | AL | BLEU |
| | 1.26 | 13.56 | 1.18 | 13.07 |
| | 1.83 | 14.98 | 1.85 | 14.67 |
| | 2.65 | 16.14 | 2.8 | 16.7 |
| Zh→En | 3.7 | 17.07 | 3.72 | 17.25 |
| | 4.26 | 17.41 | 4.54 | 17.73 |
| | 5.1 | 18 | 5.06 | 18.14 |
| | 5.71 | 18.28 | 5.85 | 18.19 |
| | 6.42 | 18.78 | 6.83 | 18.76 |
| | 7.57 | 18.97 | 8.36 | 18.88 |
| | 9.38 | 18.94 | 10.71 | 18.9 |
| | 12.22 | 18.79 | | |

**Divergence Type (Figure 8(b))**

| | Euclidean Distance | | KL Divergence | | Cosine Distance | |
|---|---|---|---|---|---|---|
| | AL | BLEU | AL | BLEU | AL | BLEU |
| | 0.67 | 11.88 | 0.72 | 12.3 | 1.18 | 13.07 |
| | 1.47 | 14.02 | 1.79 | 14.63 | 1.85 | 14.67 |
| | 2.45 | 16.34 | 2.71 | 16.36 | 2.8 | 16.7 |
| Zh→En | 3.51 | 17.11 | 4.14 | 17.45 | 3.72 | 17.25 |
| | 4.67 | 17.96 | 4.94 | 17.89 | 4.54 | 17.73 |
| | 5.53 | 18.12 | 6.32 | 18.33 | 5.06 | 18.14 |
| | 6.73 | 18.54 | 7.61 | 18.78 | 5.85 | 18.19 |
| | 8.23 | 18.74 | 8.56 | 18.83 | 6.83 | 18.76 |
| | 9.04 | 18.94 | 9.88 | 18.85 | 8.36 | 18.88 |
| | 10.1 | 18.85 | 10.8 | 18.89 | 10.71 | 18.9 |

**Number of Extra Decoder Layers (Figure 8(c))**

| | 0 Extra Decoder Layer | | 1 Extra Decoder Layer | | 3 Extra Decoder Layers | |
|---|---|---|---|---|---|---|
| | AL | BLEU | AL | BLEU | AL | BLEU |
| | 1.5 | 13.57 | 1.18 | 13.07 | 1.2 | 13.53 |
| | 2.24 | 15 | 1.85 | 14.67 | 2.29 | 15.6 |
| | 3.58 | 16.58 | 2.8 | 16.7 | 3.6 | 17.09 |
| Zh→En | 4.4 | 17.49 | 3.72 | 17.25 | 4.84 | 17.97 |
| | 6.02 | 18.3 | 4.54 | 17.73 | 6.55 | 18.67 |
| | 7.49 | 18.63 | 5.06 | 18.14 | 7.87 | 18.8 |
| | 9.7 | 18.74 | 5.85 | 18.19 | 10.05 | 18.74 |
| | | | 6.83 | 18.76 | | |
| | | | 8.36 | 18.88 | | |
| | | | 10.71 | 18.9 | | |

Table 4: Numerical results in Figure 8(a), Figure 8(b) and Figure 8(c).

| | The Maximum Number of Continuous Read (Figure 8(d)) | | | | | | | |
|---|---|---|---|---|---|---|---|---|
| | Max Conti READ 4 | | Max Conti READ 6 | | Max Conti READ 8 | | No Constraint | |
| | AL | BLEU | AL | BLEU | AL | BLEU | AL | BLEU |
| Zh→En | 0.89 | 12.45 | 1.08 | 12.74 | 1.12 | 12.76 | 1.18 | 13.07 |
| | 1.6 | 14.26 | 1.79 | 14.66 | 1.89 | 14.97 | 1.85 | 14.67 |
| | 2.36 | 15.77 | 2.7 | 16.62 | 2.83 | 16.79 | 2.8 | 16.7 |
| | 3.1 | 16.97 | 3.67 | 17.39 | 4 | 17.52 | 3.72 | 17.25 |
| | 4.25 | 17.38 | 4.44 | 17.61 | 4.9 | 17.65 | 4.54 | 17.73 |
| | 5.67 | 18.03 | 5.68 | 18.42 | 6.44 | 18.53 | 5.06 | 18.14 |
| | 7 | 18.21 | 6.87 | 18.7 | 7.74 | 18.48 | 5.85 | 18.19 |
| | 9.12 | 18.35 | 8.57 | 18.62 | 9.73 | 18.78 | 6.83 | 18.76 |
| | | | 10.84 | 19.14 | 12.06 | 19.19 | 8.36 | 18.88 |
| | | | | | | | 10.71 | 18.9 |
| | Max Conti READ 4 | | Max Conti READ 6 | | Max Conti READ 8 | | No Constraint | |
| | AL | BLEU | AL | BLEU | AL | BLEU | AL | BLEU |
| De→En | 0.49 | 21.65 | 0.89 | 22.56 | 1.13 | 23 | 1.79 | 23.88 |
| | 1.3 | 24.51 | 1.76 | 25.39 | 2.04 | 25.99 | 2.5 | 26.07 |
| | 2.17 | 27.12 | 2.69 | 28.01 | 3.09 | 28.45 | 3.24 | 27.81 |
| | 3.25 | 29.19 | 3.54 | 29.39 | 3.98 | 29.77 | 4.03 | 29.12 |
| | 4.31 | 29.97 | 4.53 | 30.23 | 5.16 | 30.54 | 5.13 | 30.22 |
| | 5.87 | 30.84 | 5.55 | 30.83 | 6.32 | 31.12 | 6.02 | 30.77 |
| | 7.65 | 31.29 | 6.36 | 31.12 | 7.25 | 31.34 | 7.59 | 31.05 |
| | 8.98 | 31.52 | 7.5 | 31.54 | 8.59 | 31.66 | 8.88 | 31.39 |
| | 10.53 | 31.6 | 9.77 | 31.7 | 11.21 | 31.89 | 11.13 | 31.56 |
| | 12.53 | 31.79 | | | | | | |
| | Max Conti READ 4 | | Max Conti READ 6 | | Max Conti READ 8 | | No Constraint | |
| | AL | BLEU | AL | BLEU | AL | BLEU | AL | BLEU |
| En→Vi | 0.88 | 21.81 | 0.89 | 21.87 | 0.89 | 21.89 | 0.89 | 21.89 |
| | 1.4 | 27.03 | 1.41 | 27.11 | 1.41 | 27.12 | 1.41 | 27.11 |
| | 1.94 | 29.3 | 1.97 | 29.27 | 1.98 | 29.3 | 1.99 | 29.31 |
| | 2.74 | 29.64 | 2.89 | 29.56 | 2.96 | 29.58 | 3.06 | 29.63 |
| | 3.7 | 30.04 | 4.08 | 30.02 | 4.31 | 30.07 | 4.6 | 30.15 |
| | 4.24 | 30.01 | 4.76 | 29.98 | 5.05 | 30.02 | 5.44 | 30.09 |
| | 5.38 | 30.07 | 5.32 | 30.07 | 5.7 | 30.08 | 6.25 | 30.13 |
| | 6.1 | 30.2 | 6.21 | 30.09 | 6.69 | 30.05 | 7.49 | 30.15 |
| | 7.15 | 30.04 | 7.1 | 30.18 | 7.7 | 30.19 | 8.08 | 30.2 |
| | 7.76 | 30.12 | 7.67 | 30.08 | 8.32 | 30.04 | 8.74 | 30.17 |
| | 8.49 | 30.17 | 8.35 | 29.98 | 9.06 | 30.04 | 9.61 | 30.01 |
| | 11.04 | 30.03 | 9.04 | 30.02 | 9.83 | 30.06 | 10.67 | 30.11 |
| | | | 9.9 | 30.01 | 10.83 | 30.02 | 11.69 | 30.1 |
| | | | 12.77 | 30.05 | | | | |

Table 5: Numerical results in Figure 8(d).