# OpenReview forum: "Adaptive Policy with Wait-k Model for Simultaneous Translation"
_EMNLP/2023/Conference — EMNLP 2023 Main_

### Official Review · Reviewer_sUJ2 · 2023-07-27

**Soundness:** 4

**Excitement:**

3: Ambivalent: It has merits (e.g., it reports state-of-the-art results, the idea is nice), but there are key weaknesses (e.g., it describes incremental work), and it can significantly benefit from another round of revision. However, I won't object to accepting it if my co-reviewers champion it.

**Missing References:**

-	ACL 2023: Learning Optimal Policy for Simultaneous Machine Translation via Binary Search
	- This work also observes that the generation probability of the target word is related to the amount of source information, and uses binary search to find the optimal policy based on this, and builds a separate module to learn the policy.

-	Some of the latest SIMT work has achieved similar performance to ITST, which can be referred to.
	-	ICLR 2023: Hidden Markov Transformer for Simultaneous Machine Translation
	-	ACL 2023: Better Simultaneous Translation with Monotonic Knowledge Distillation


**Paper Topic And Main Contributions:**

In this study on Simultaneous Machine Translation (SiMT), the researchers propose a flexible approach by decoupling the adaptive policy model from the translation model. They introduce DaP, a divergence-based adaptive policy, which efficiently makes read/write decisions based on prediction discrepancy. DaP extends a frozen wait-k model with lightweight parameters for adaptive policy. Experimental results across show that their approach improves translation accuracy and latency trade-off.

The proposed method outperforms the current state-of-the-art ITST, and the policy is somewhat interpretable. The article is well written and easy to understand.


**Questions For The Authors:**

Q1: Have you tried jointly fine-tuning Multi-path Wait-k Model and R/W Policy Net to enhance the performance of the translation model on the predicted READ/WRITE path from Policy Net? Can freezing the Multi-path Wait-k Model achieve comparable performance to fine-tuning?

Q2: Since the R/W Policy Net is the main part of the proposed method, I would like to know how accurate the R/W Policy Net is in predicting divergence. Experiments on R/W Policy Net prediction accuracy and READ/WRITE path should be added.

Q3: The effectiveness of DaP-SiMT strongly depends on the correctness and generalization of the empirical observations of divergence matrix. Therefore, can you give some statistical properties about the divergence matrix (such as correlation with alignment) instead of some case studies.


Q4: Considering that the divergence matrix is calculated by the offline MT model, can the offline model be used instead of the Multi-path Wait-k Model in the proposed method? Because the lower divergence indicates that offline MT predicts that the distribution of the current target word is similar based on full-sentence and prefix, can offline MT be used directly?

The issue of whether the translation model uses offline, frozen Multi-path Wait-k, or trainable Multi-path Wait-k should be systematically analyzed, which will help to enhance the universality of the proposed method.


**Reasons To Accept:**

-	SiMT has received more and more attention due to its low latency characteristics. The proposed method achieves promising results, surpassing the current state-of-the-art ITST.
-	The proposed divergence-based policy is interesting and has certain interpretability.
-	The proposed method adopts a frozen multi-path wait-k translation model and only needs to train the policy module.
-	The article is well written and easy to follow.


**Reasons To Reject:**

-	The article has no obvious weaknesses to reject.
-	A potential issue is that divergence-based policies may be affected by language pairs. For example, the word order difference in De-En may make the model READ more words, and the author added Max-Continuous-READ to alleviate this issue.


**Reproducibility:**

4: Could mostly reproduce the results, but there may be some variation because of sample variance or minor variations in their interpretation of the protocol or method.

**Reviewer Confidence:**

5: Positive that my evaluation is correct. I read the paper very carefully and I am very familiar with related work.

**Typos Grammar Style And Presentation Improvements:**

-	Section 4.3 should add some formulas to clearly explain the training objective of R/W Policy Net. Also, the "READ or WRITE" in Figure 5 is confusing. In my understanding, R/W Policy Net learns each item in the divergence matrix instead of directly learning READ/WRITE. It is recommended that you provide a clearer explanation in the final version.

---

> ### Author Rebuttal · Authors · 2023-08-29
>
> ### Thanks for the valuable comments. Here is the response to Reviewer sUJ2
>
> > Q1: Have you tried jointly fine-tuning Multi-path Wait-k Model and R/W Policy Net to enhance the performance of the translation model on the predicted READ/WRITE path from Policy Net? Can freezing the Multi-path Wait-k Model achieve comparable performance to fine-tuning?
>
> A1: We appreciate this professional suggestion. As we mentioned the motivation to decouple the MT model and RW policy net, which is more applicable in practice, so we did not explore the joint fine-tuning.
>
> In order to address this concern, we conducted a simple experiment on jointly fine-tuning. For quick and stable training, we initialized the model with our well-trained DaP-SiMT checkpoint. To optimize the model, we apply the original multi-path wait-k loss, the sequence cross-entropy loss with the derived path based on our RW policy net, and the RW regression loss.
>
> In this setup, we observe that the AL-BLEU curve of jointly fine-tuning is almost identical to the model by freezing Multi-path Wait-k Model. Further exploration in this direction will be our future work.
>
> > Q2: Since the R/W Policy Net is the main part of the proposed method, I would like to know how accurate the R/W Policy Net is in predicting divergence. Experiments on R/W Policy Net prediction accuracy and READ/WRITE path should be added.
>
> A2: Thanks for your expertise and attention to our work. First, we would address that it is almost impossible to directly compare the predicted divergence matrix and the corresponding ground truth or calculate the accuracy, because the predicted target has different lengths and different tokens in the same position with the reference. So in our paper, we propose a pseudo-predicted divergence matrix in the case study, where the target used is the reference rather than predicted result.
>
> Second, to be honest, it is difficult for the RW policy net to predict accurate divergence values at every single position in the matrix. As shown in the case study, the absolute differences for the same positions are non-negligible. Interestingly, we observe that the predicted divergence values effectively learn the relative magnitude relationship, which could result in a similar path as the ground truth matrix if an appropriate threshold is selected.
>
> Therefore, our main concern is whether suitable R/W paths can be obtained based on the predicting matrix. To achieve this goal, we analyze the quality of the path with the proposed AL-NLL metric in Section 5.4.1.
>
> For this response, we came up with a method for accuracy evaluation on two R/W paths. Specifically, we align two paths on the same matrix and calculate the area surrounded by two paths. Then the accuracy is $1 - \frac{S_{area}}{|x| |y|}$. If the predicted path completely coincides with the ground truth path, the area surrounded is 0, then the accuracy is 1. We calculated the accuracy on the Zh-En test set at different AL values. The table below shows the results.
>
> |  |  |  |  |  |  |  |  |  |  |  |  |  |
> | --- | --- | --- | --- | --- | --- | --- | --- | --- | --- | --- | --- | --- |
> |  | AL | 0.55 | 1.51 | 2.46 | 3.41 | 4.32 | 5.23 | 6.12 | 7.83 | 8.55 | 9.98 | 10.69 |
> | Predicted Path  | Accuray | 0.898 | 0.915 | 0.913 | 0.909 | 0.9 | 0.895 | 0.893 | 0.894 | 0.897 | 0.906 | 0.911 |
> | Wait-k Path  | Accuray | 0.85 | 0.857 | 0.845 | 0.833 | 0.822 | 0.815 | 0.814 | 0.816 | 0.82 | 0.83 | 0.835 |
>
> > Q3:  The effectiveness of DaP-SiMT strongly depends on the correctness and generalization of the empirical observations of the divergence matrix. Therefore, can you give some statistical properties about the divergence matrix (such as correlation with alignment) instead of some case studies.
>
> A3: This is an insightful question. We have also carefully considered to quantify the effectiveness of the divergence matrix with the proposed AL-NLL metric in Section 5.4.1.
>
> In fact, the turning point of a suitable R/W path may not necessarily indicate an alignment. However, when preparing for a write operation, the current source prefix should usually include the source tokens aligned to the target prefix, which means that a good R/W path should have less hallucination or a smaller Anticipation Rate (defined in Appendix B). Therefore, we have also added the AR results of the R/W path obtained based on the divergence matrix in the Zh-En test set.
>
> |  |  |  |  |  |  |  |  |  |  |  |  |
> | --- | --- | --- | --- | --- | --- | --- | --- | --- | --- | --- | --- |
> |  | AL | 1.51 | 2.46 | 3.41 | 4.33 | 5.23 | 6.12 | 7.83 | 8.55 | 9.38 | 10.69 |
> | Divergence Matrix based Path | AR | 28.06 | 18.60 | 12.52 | 8.84 | 6.38 | 4.74 | 2.67 | 2.17 | 1.84 | 1.25 |
> | Wait-k Policy based Path | AR | 31.40 | 23.37 | 17.54 | 13.33 | 10.26 | 7.85 | 4.79 | 3.73 | 3.00 | 1.81 |
>
> More evidence to show the effectiveness or potential of our approach can also refer to the response to Question A for Reviewer Pp2p.
>
> > Q4: Considering that the divergence matrix is calculated by the offline MT model, can the offline model be used instead of the Multi-path Wait-k Model in the proposed method? Because the lower divergence indicates that offline MT predicts that the distribution of the current target word is similar based on full-sentence and prefix, can offline MT be used directly?
>
> A4: Theoretically, offline MT is not suitable for the streaming input in simultaneous translation, because it always requires bidirectional attention and full context. However, we would like to follow the reviewer's suggestion and test the SiMT performance based on the offline MT in the Zh-En test dataset.  The results are shown in the following table. As expected, the BLEU score of the offline MT model is higher in the high-latency region. However, in the mid-to-low latency range, the translation performance of the offline MT model is not ideal. Additional Prefix2Prefix training (as the joint fine-tuning mode in Q1) for the offline MT model should improve this situation.
>
> - DaP-SiMT based on Wait-k model
> |  |  |  |  |  |  |  |  |  |  |  |
> | --- | --- | --- | --- | --- | --- | --- | --- | --- | --- | --- |
> | AL | 0.96 | 1.61 | 2.36 | 2.95 | 3.74 | 4.67 | 5.33 | 6.36 | 7.88 | 10.06 |
> | BLEU | 12.55 | 14.23 | 16.08 | 16.92 | 17.72 | 17.83 | 18.09 | 18.65 | 18.62 | 19.16 |
>
> - DaP-SiMT based on offline MT model
> |  |  |  |  |  |  |  |  |  |  |
> | --- | --- | --- | --- | --- | --- | --- | --- | --- | --- |
> | AL | 1.52 | 2.67 | 3.28 | 4.02 | 4.78 | 5.64 | 6.93 | 7.91 | 9.41 |
> | BLEU | 9.51 | 13.67 | 15.38 | 16.28 | 17.03 | 17.92 | 18.11 | 18.73 | 19.32 |
>
> **Missing References**
>
> Thank you for your reminder. We will include these references in the final version of the paper.
>
> **Typos Grammar Style And Presentation Improvements**
>
> Your understanding is completely correct, and we appreciate your suggested revisions. We will make improvements based on your recommendations in the final version of the paper.

---

### Official Review · Reviewer_Pp2p · 2023-08-04

**Soundness:** 4

**Excitement:**

4: Strong: This paper deepens the understanding of some phenomenon or lowers the barriers to an existing research direction.

**Missing References:**

I have not detected missing references

**Paper Topic And Main Contributions:**

This paper proposes an adaptive policy net for SiMT that is trained independently from the translation model and based on the idea of divergence between the prefix-based and the full-sentence probability distro. This adaptive policy is integrated with the multi-path SiMT model (but could be other SiMT model) to prove that outperforms the SoTA ITST SiMT model. This paper provides an alternative simpler approach to policy training in SiMT that is supported by extensive experiments with positive results and ablation studies.

**Questions For The Authors:**

Question A: Have you computed the curves for "DaP-SiMT Ground Truth" for Figure 6? I would like to see the gap between the prediction and the ground truth in terms of BLEU vs. AL.

Question B: I have read in the ablation studies that there seem not to be an effect of the divergence type, but have you tried other divergences such as Jensen-Shannon?

Question C: I guess you have determined the thresholds for Euclidean, KL-divergence and cosine distance using the validation sets of each task, but could you elaborate on that? How sensitive are BLEU vs. AL to these thresholds? What type of divergence do you use for Figure 6?

**Reasons To Accept:**

The approach to independently train the adaptive policy based on the idea of divergence is original and the results support the benefits of this simpler approach. The presentation of the paper is excellent and easy to understand the motivation behind the authors' decisions.

**Reasons To Reject:**

The translation quality improvement over the SoTA ITST model is not significant, but systematic across tasks and latency values.

**Reproducibility:**

4: Could mostly reproduce the results, but there may be some variation because of sample variance or minor variations in their interpretation of the protocol or method.

**Reviewer Confidence:**

4: Quite sure. I tried to check the important points carefully. It's unlikely, though conceivable, that I missed something that should affect my ratings.

**Typos Grammar Style And Presentation Improvements:**

Typo:
- Table 3 : Ectra -> Extra

Presentation improvements:
- Tables should have caption as header.
- Ground truth curves for Figure 6.

---

> ### Author Rebuttal · Authors · 2023-08-29
>
> ### Thanks for the insightful comments. We will address your concern and questions accordingly.
>
> > Concern: The translation quality improvement over the SoTA ITST model is not significant, but systematic across tasks and latency values.
>
> Thanks for raising this issue. We will make it more clear in the revised version.
>
> In Figure 6 of our paper's main experimental results, it visually appears that the two curves (our DaP-SiMT and ITST) are close. However, when we focus on the AL-BLEU curves in the low-latency region and observe the BLEU gap for given latency, we can see that our method is actually much superior to ITST, e.g., when AL$\approx$2, $\Delta$BLEU$\approx +1.5$ for Zh-En,  $\Delta$BLEU$\approx +2.5$ for De-En, $\Delta$BLEU$\approx +3$ for En-Vi. Because the curve is steep in the low-latency region, the actual BLEU difference is larger than it visually appears.
>
> > Question A: Have you computed the curves for "DaP-SiMT Ground Truth" for Figure 6? I would like to see the gap between the prediction and the ground truth in terms of BLEU vs. AL.
>
> Thanks for your professional suggestions. We suppose your suggestion is to calculate the R/W signal by $D(p_{pre}^t, p_{full}^t)$ in Eqn. (6,7) during inference, while the target generation remains the same $p(y_t | y_{<t}, x_{\leq g(t)})$.
>
> We have computed the curves for "DaP-SiMT Ground Truth" for Figure 6 on the Zh-En test set (as shown in the table below) and will include this result in the final version of the paper. As expected, the results of DaP-SiMT Ground Truth guided inference are significantly better ($\Delta$BLEU$\approx +1.7$ for AL$\approx 2.3$) and can be considered as the upper bound of our current approach. In future work, we can continuously approach this upper bound by optimizing the regression task of the RW policy net.
>
> - DaP-SiMT Ground Truth guided inference
> |  |  |  |  |  |  |  |  |  |
> | --- | --- | --- | --- | --- | --- | --- | --- | --- |
> | AL | 0.41 | 1.02 | 1.77 | 2.28 | 2.91 | 3.56 | 4.71 | 6.19 |
> | BLEU | 13.46 | 14.84 | 17.13 | 17.77 | 18.26 | 18.29 | 18.81 | 18.94 |
>
> - DaP-SiMT
> |  |  |  |  |  |  |  |  |  |  |  |
> | --- | --- | --- | --- | --- | --- | --- | --- | --- | --- | --- |
> | AL | 0.96 | 1.61 | 2.36 | 2.95 | 3.74 | 4.67 | 5.33 | 6.36 | 7.88 | 10.06 |
> | BLEU | 12.55 | 14.23 | 16.08 | 16.92 | 17.72 | 17.83 | 18.09 | 18.65 | 18.62 | 19.16 |
>
> > Question B:  I have read in the ablation studies that there seem not to be an effect of the divergence type, but have you tried other divergences such as Jensen-Shannon?
>
> Thanks for your expertise in our field. We conducted experiments based on Jensen-Shannon divergence (JSD) and found that its AL-BLEU curve is comparable to other divergences mentioned in the paper (with slightly lower BLEU scores in the low-latency region). However, we do not recommend using JSD because it has a more concentrated distribution compared to other divergences. This concentration makes the determination of an appropriate threshold for the specific AL more sensitive (as shown in the table below), resulting in difficulties in finding a suitable threshold.
>
> |  |  |  |  |  |  |  |  |  |  |  |  |
> | --- | --- | --- | --- | --- | --- | --- | --- | --- | --- | --- | --- |
> | AL | 1.44 | 2.7 | 3.68 | 4.52 | 5.85 | 6.41 | 7.88 | 8.92 | 9.76 | 10.69 | 11.2 |
> | threshold | 0.26 | 0.14 | 0.1 | 0.08 | 0.06 | 0.055 | 0.045 | 0.04 | 0.036 | 0.032 | 0.030 |
>
> > Question C: I guess you have determined the thresholds for Euclidean, KL-divergence, and cosine distance using the validation sets of each task, but could you elaborate on that? How sensitive are BLEU vs. AL to these thresholds? What type of divergence do you use for Figure 6?
>
> Thanks for your reminder. We will include relevant discussion in the revised version.
>
> As you mentioned, we first determined the list of thresholds on the validation set and then applied them to the test set. The detailed process is similar to find the threshold list in ROC curve.
>
> The specific approach is a modified grid search method, where we start by testing the AL and BLEU values with thresholds of {0.1, 0.2, ..., 1.0} with the interval of 0.1. If some regions (e.g., [0.1, 0.3]) have a large drop for AL, we continue to search within such regions with a smaller interval (such as 0.02) until finding a suitable threshold. Alternatively, a simple and native approach is to directly test BLEU-AL on all thresholds with a predefined minimum interval, e.g., 0.02.
>
> Because the AL is not particularly sensitive to the threshold overall, the process of determining the threshold is straightforward. The table below shows the relationship between AL and threshold for Cosine distance, Euclidean, and KL-divergence, respectively. The divergence used in Figure 6 is Cosine distance.
>
> - Cosine distance
> |  |  |  |  |  |  |  |  |  |  |
> | --- | --- | --- | --- | --- | --- | --- | --- | --- | --- |
> | AL | 1.18 | 1.85 | 2.8 | 3.72 | 4.54 | 5.85 | 6.83 | 8.36 | 10.71 |
> | threshold | 0.52 | 0.4 | 0.26 | 0.18 | 0.14 | 0.1 | 0.08 | 0.06 | 0.04 |
>
> - Euclidean distance
> |  |  |  |  |  |  |  |  |  |  |  |  |
> | --- | --- | --- | --- | --- | --- | --- | --- | --- | --- | --- | --- |
> | AL | 0.67 | 1.47 | 2.1 | 3.51 | 4.67 | 5.53 | 6.73 | 7.42 | 9.04 | 10.01 | 11.19 |
> | threshold | 0.5 | 0.4 | 0.33 | 0.24 | 0.2 | 0.18 | 0.16 | 0.15 | 0.13 | 0.12 | 0.11 |
>
> - KL-divergence
> |  |  |  |  |  |  |  |  |  |  |  |  |
> | --- | --- | --- | --- | --- | --- | --- | --- | --- | --- | --- | --- |
> | AL | 0.72 | 1.37 | 2.21 | 3.03 | 4.07 | 4.94 | 6.32 | 7.61 | 8.56 | 9.88 | 10.8 |
> | threshold | 1.8 | 1.4 | 1.0 | 0.7 | 0.5 | 0.4 | 0.3 | 0.24 | 0.2 | 0.16 | 0.14 |
>
> **Typos Grammar Style And Presentation Improvements**
>
> Thank you for your constructive suggestions, and we will make improvements based on your recommendations in the final version of the paper.

---

### Official Review · Reviewer_VTic · 2023-08-06

**Soundness:** 4

**Excitement:**

4: Strong: This paper deepens the understanding of some phenomenon or lowers the barriers to an existing research direction.

**Missing References:**

1. Zheng et al. Simpler and Faster Learning of Adaptive Policies for Simultaneous Translation. EMNLP, November 2019.
2. Zheng et al. Simultaneous translation policies: from fixed to adaptive. ACL, 2020, arXiv

**Paper Topic And Main Contributions:**

This paper is about the development of an adaptive policy with wait-k model for simultaneous translation (SiMT). The paper addresses the challenge of SiMT, which requires a trade-off between translation accuracy and latency. One of the contributions of the paper is the proposal of a flexible approach to SiMT by decoupling the adaptive policy model from the translation model. This approach allows for improved trade-off between translation accuracy and latency. The paper also proposes a new read/write supervision signal for the adaptive policy model, which is based on the wait-k model. The experimental results show that the proposed approach outperforms strong baselines in terms of both translation accuracy and latency.

**Questions For The Authors:**

Could you please compare your method with the following two papers?
1. Zheng et al. Simpler and Faster Learning of Adaptive Policies for Simultaneous Translation. EMNLP, November 2019.
2. Zheng et al. Simultaneous translation policies: from fixed to adaptive. ACL, 2020, arXiv

**Reasons To Accept:**

The main contribution of the paper is the proposal of a novel method to construct read/write supervision signals from a parallel training corpus based on statistical divergence. The paper also presents a lightweight policy model that enables adaptive read/write decision-making for a well-trained multi-path wait-k translation model. The proposed approach is both memory and computation efficient and offers an improved trade-off between translation accuracy and latency, outperforming strong baselines. The paper makes contributions to the field of NLP engineering experiment and provides a new approach for SiMT that can be applied to various languages.

**Reasons To Reject:**

There are two papers which are very related to this paper:
1. Zheng et al. Simpler and Faster Learning of Adaptive Policies for Simultaneous Translation. EMNLP, November 2019.
2. Zheng et al. Simultaneous translation policies: from fixed to adaptive. ACL, 2020, arXiv

The authors should compare their work with these two lines of work.

**Reproducibility:**

4: Could mostly reproduce the results, but there may be some variation because of sample variance or minor variations in their interpretation of the protocol or method.

**Reviewer Confidence:**

4: Quite sure. I tried to check the important points carefully. It's unlikely, though conceivable, that I missed something that should affect my ratings.

---

> ### Author Rebuttal · Authors · 2023-08-29
>
> ### Thanks for the valuable comments. Here is our response.
>
> > Q1: There are two papers which are very related to this paper. Could you please compare your method with the following two papers?
>
> [1] Zheng et al. Simpler and Faster Learning of Adaptive Policies for Simultaneous Translation. EMNLP, November 2019.
>
> [2] Zheng et al. Simultaneous translation policies: from fixed to adaptive. ACL, 2020, arXiv
>
> A1: Thanks for the professional suggestion. The two papers are excellent in the field of simultaneous translation. We will cite both of these papers in the final version of our paper. However, since these papers did not have open-source code and our experimental settings differ from them, it is challenging to achieve a fair comparison. Nevertheless, we have found that the ITST paper reproduced the method [2] with the same experimental setting as ours in the De-En scenario. We therefore include a comparison in this specific experimental setting as follows. It shows that our DaP-SiMT outperforms the method in [2].
>
> - Adaptive Wait-k [2]
> |  |  |  |  |  |  |  |  |  |  |  |  |
> | --- | --- | --- | --- | --- | --- | --- | --- | --- | --- | --- | --- |
> | AL | 0.83 | 1.9 | 2.94 | 4.10 | 5.11 | 6.09 | 7.21 | 8.23 | 10.12 | 11.55 | 12.18 |
> | BLEU | 20.29 | 23.56 | 25.96 | 27.44 | 28.29 | 28.91 | 29.73 | 30.10 | 30.76 | 30.78 | 30.74 |
>
> - DaP-SiMT
> |  |  |  |  |  |  |  |  |  |  |  |
> | --- | --- | --- | --- | --- | --- | --- | --- | --- | --- | --- |
> | AL | 0.49 | 1.3 | 2.17 | 3.25 | 4.31 | 5.87 | 7.65 | 8.98 | 10.53 | 12.53 |
> | BLEU | 21.65 | 24.51 | 27.12 | 29.19 | 29.97 | 30.84 | 31.29 | 31.52 | 31.6 | 31.79 |

---

### Meta-Review · Area_Chair_UDtN · 2023-09-07

**Recommendation:** 5

**Metareview:**

The paper moves the pareto frontier for the quality-latency tradeoff in simultaneous machine translation by proposing to separate the adaptive policy model from the translation model and introducing a supervision signal based on statistical divergence.

The reviewers agree that this work is technically sound, well written, and exciting for the community.

---

### Decision · Program_Chairs · 2023-10-07

**Decision:**

Accept-Main

**Comment:**

The paper moves the pareto frontier for the quality-latency tradeoff in simultaneous machine translation by proposing to separate the adaptive policy model from the translation model and introducing a supervision signal based on statistical divergence.

The reviewers agree that this work is technically sound, well written, and exciting for the community.